# Genomic Survey of Genes Encoding Major Intrinsic Proteins (MIPs) and Their Response to Arsenite Stress in Pepper (*Capsicum annum*)

**DOI:** 10.3390/plants14101475

**Published:** 2025-05-14

**Authors:** Syed Muhammad Azam, Kaixuan Huang, Jiaxin Yuan, Yanqing Bai, Qiaolin Chen, Panpan Dang, Hend Alwathnani, Hajar Fahad Bin Zayid, Renwei Feng, Christopher Rensing

**Affiliations:** 1Institute of Environmental Microbiology, College of Resources and Environment, Fujian Agriculture & Forestry University, Fuzhou 350002, China; syedazamfafu@gmail.com (S.M.A.); 13388750474@163.com (Y.B.);; 2Department of Botany and Microbiology, King Saud University, Riyadh 11495, Saudi Arabia; wathnani@ksu.edu.sa (H.A.); 444203445@student.ksu.edu.sa (H.F.B.Z.)

**Keywords:** aquaporin, heavy metal, major intrinsic protein, arsenite, abiotic stress

## Abstract

Major intrinsic proteins (MIPs) are a super family of proteins that mediate the bidirectional concentration-dependent flux of water in particularly small solutes in fraction and some metalloids across the cell membrane. This article reports the genome-wide study of pepper genes encoding MIPs and their expression analysis. Using a bioinformatics homology search, 48 CAMIPs were identified on the genome of pepper. A total of 48 MIPs were further divided in sub classes as 22 CATIPs, 15 CAPIPs, 10 CANIPs, and 1 CASIP. The 48 Pepper MIP encoding genes were mapped on the 12 pepper chromosomes. CAMIP synteny analysis exhibited 17 duplicated genes, and these were clustered into eight tandem duplicated regions on pepper chromosomes. The tissue-specific expression of MIPs based on RNA-Seq showed certain CANIPs, CATIPs, and CAPIPs were highly expressed in roots, while some CATIPs and CASIPs were expressed in stem as well. As(III), at 0.5 and 1 mM, was applied to pepper plants, where 1 mM significantly reduced leaf chlorophyll content, leaf nitrogen content, and root length. To see which CAMIPs participate in As(III) transport, we tested the response of genes encoding MIPs to As(III) through qRT-PCR. As(III) uptake was observed in both shoot and root samples treated with 0.5 mM and 1 mM As(III) for 12 h and 24 h because of MIPs’ quantitative response through qRT-PCR. Most of the MIPs were down-regulated in response to both levels of As(III); besides CANIPs, there were CATIPs and CAPIPs up-regulated in response to higher concentrations of As(III) in the roots and shoot, which suggests the involvement of CAMIPs in the uptake as well as detoxification mechanism in pepper against As(III). Unlike prokaryotes, plant MIPs have diverse selectivity for arsenite and other solutes. Our study provides important insights into the arsenite uptake and detoxification, offering a foundation for further functional and stress-tolerance studies.

## 1. Introduction

Cells are defined by their ability to reproduce, grow, respond to external stimuli, and perform different biological processes. The cell membrane acts as a selective barrier which controls the movement of substances in or out of the cell. Major intrinsic proteins (MIPs) are a super family of constitutional membrane proteins that facilitate the bidirectional movement of water and small solutes across membranes. MIPs have been extensively investigated in nearly all organisms. The MIP superfamily contains two main evolutionary related distinct groups; water-specific channels (aquaporins), and the water, glycerol, and other small, uncharged solute-transporting channels (aquaglyceroporins) [1]. Aquaporins and aquaglyceroporins display a highly conserved structure, consisting of six transmembrane helices; which are connected by five loops, named Loop A to Loop E, with two conserved asparagine–proline–alanine (NPA) motifs disseminated in Loop B and Loop E [2]. Furthermore, there are aromatic/arginine (Ar/R) selectivity filters, including residues of Helix 2, Helix 5, Loop E1, and Loop E2, and Froger’s positions, containing residues of P1 to P5 [3]. Plant MIPs show a higher degree of diversity and therefore cannot be clearly differentiated into the aquaporins and aquaglyceroporins, because of variable solute selectivity.

A member of the MIP family, the glycerol facilitator (*GlpF*) from *Escherichia coli*, was the first one identified, characterized, and shown to transport water and glycerol and therefore termed aquaglyceroporin, while *AQP1* was the first identified in humans. After the discovery of the first aquaporin, MIPs have already been identified in a wide range of other organisms, including Archaea [4], protozoa [5], bacteria [6], liver [7], insects [8] and plants [9,10,11]. In plants, members of the MIP family are especially prevalent, with over 30 genes encoding MIPs typically found on the higher plant genome [12]. Higher plants have more aquaporin isoforms compared to other kingdoms. There are 35 genes encoding MIPs in the model plant *Arabidopsis thaliana*, while 33 genes encoding MIPs have been reported in *Oryza sativa*. Although evolutionary differences exists between different aquaporins, the typical protein structure was observed to be highly conserved with NPA motif (Asp-Pro-Ala) tetrameric quaternary structure characterized by a central conserved aqueous pore [13,14]. In the context of metal(oid) transport via MIPs, a mutant of *GlpF* was first isolated as being resistant to antimonite Sb(III) [15].

Phylogenetically, the encoded MIPs have been further divided into four subfamilies: small basic intrinsic proteins (SIPs), nodulin 26-like intrinsic proteins (NIPs), plasma membrane intrinsic proteins (PIPs) and tonoplast intrinsic proteins (TIPs). The main biochemical factors associated with plant–water relations include the “aquaporins”, being 26–30 kDa water channel proteins. These proteins were shown to specifically mediate the passive transport of water molecules across membranes, thereby regulating the trans-cellular route of water [16,17].

A mutant of the glycerol facilitator of *Escherichia coli* also caused a 90% reduction in arsenite (As(III)) uptake [18]. *Fps1* is the homolog of *GlpF* in *Saccharomyces cerevisiae* that was reported to mediate the influx of As(III) and Sb(III) [1]. Members of the NIPs and PIPs have been identified to facilitate As(III) transport in plants in addition to silicon and boron transport as essential nutrients [19,20]. NIPs are permeable to glycerol, ammonia, urea, lactic acid, formamide, and metalloids [1].

Arsenic is abundant in the Earth’s crust and has been shown to enter the food chain from drinking water that has flowed or flooded through arsenic-containing soil. Arsenic is toxic even in small amounts. Arsenic, a group 1 human carcinogen, is the most abundant environmental toxin and carcinogen, according to the U.S. Environmental Protection Agency (EPA) (http://www.atsdr.cdc.gov/SPL/index.html, accessed on 1 November 2024). Since Sb(III) is a sister metalloid of As(III) (generally called sister metals because of their homology), the affected root morphology in various reported plant species and the changes in root morphology were shown to be correlated to changes in root cell-wall-related enzymes. Arsenic and antimony are non-essential toxic metalloids and their high concentration in rice grains and effects on root morphology result in a serious threat to rice yield and quality, as well as a serious threat to human health [21,22]. During a study that was conducted in *Brassica napus* and *Brassica juncea*, using 25 to 75 mg As kg^−1^, a significant decrease in growth parameters such as number of leaves, leaf area, plant height, and shoot and root dry biomass could be observed. Gas exchange traits (transpiration rate, photosynthetic rate, stomatal conductance) exhibited a decrease, and photosynthetic pigments and water-use efficiency were also adversely affected [23]. Here, we conducted an experiment exploring MIP encoding genes using bioinformatics and biostatistics on the pepper genome. Publicly available RNA-Seq was used to assess tissue-specific expression profiles of CAMIPs. Furthermore, root length and leaf chlorophyll and leaf nitrogen contents were also calculated after As(III) treatment, followed by observing the expression profiles of CAMIPs in response to arsenite stress through qRT-PCR.

## 2. Results

### 2.1. Identification and Classification of Genes Encoding MIPs in Pepper

To determine MIP family homologs in pepper, a genome-wide search was performed using BLAST searches with 34 MIP genes from rice and 35 MIP genes from Arabidopsis as queries. A total of 48 non-redundant MIP genes were revealed on the pepper genome. Furthermore, 48 MIPs were categorized into four subgroups, including 15 PIPs (plasma membrane intrinsic proteins), 22 TIPs (tonoplast intrinsic proteins), 10 NIPs (NOD26-like intrinsic proteins) and 1 small basic intrinsic protein (SIP) (Table 1).

### 2.2. MIP Genes Characteristics and Chromosomal Localization

Pepper MIP genes were characterized and explored in more detail. The molecular weight of MIP genes ranged from 4950.4 to 35,282.6 kd; *CAPIP1-7* exhibited a higher molecular weight, while *CATIP1-12* had a lower molecular weight. The number of A.A varied from 107 for *CATIP2-1* to 329 A.A for *CANIP5-1*. The isoelectric point (Ip) is the pH at which a molecule has no net electric charge. A minimum Ip of (4.2) was observed for *CATIP1-10*, while a maximum Ip (9.8) was noted for *CATIP1-9*. The number of ORFs were also identified, which varied from 2 to 10 for *CATIP1-10* and *CAPIP2-2*, respectively. The instability index tells us about protein’s stability in the test tube. If the instability index is less than 40, then it is regarded as stable in the test tube, while a greater instability index indicates it is probably not a stable protein. Here, MIP genes exhibited instability indices lower than 40 for most of the MIP gene products, indicating most pepper MIPs are stable. The proportional volume occupied by the aliphatic side chains (valine, leucine, alanine, and isoleucine) in a protein is regarded as the aliphatic index. It is regarded as a positive factor for the increase in thermostability of orbicular proteins. The aliphatic index for pepper MIPs was minimum (82.23) for *CANIP4-1*, while it was maximum (140.18) for *CATIP1-2* (Table 1). MIPs were distributed on their respective location on the chromosomes. Among the 48 MIP genes in pepper there were 6 genes (*CATIP1-12*, *CATIP2-2*, *CAPIP1-3*, *CAPIP2-1*, *CANIP5-1* and *CANIP4-1*) that were not assigned to any chromosomes, so they were placed on the unplaced Scaffold, while the other 42 genes were distributed on all 12 chromosomes. Chromosomes 1 and 6 have the maximum number of MIP genes, seven CAMIP encoding genes, followed by chromosome 11, which has six CAMIPs located on it (Table 1, Figure 1).

### 2.3. Phylogeny, Gene Structure and Motif Structure of MIP Genes

Phylogenetic relationships were elucidated between 35 Arabidopsis, 34 rice, and 48 pepper MIP-encoded genes; a three-species unrooted mixed phylogenetic tree was designed using MEGA6.0, and the Neighbour-Joining method was carried out. The bootstrap test was performed with 1000 iterations founded on the alignment of the MIP domain sequences (Figure 2). Pepper MIPs homologs were divided into four groups, i.e., CAPIPs, CATIPs, CANIPs and CASIPs. There were 15 genes in CAPIPs, 22 genes in CATIPs, 10 genes in CANIPs, and 1 gene in CASIP (Figure 2), while the *Arabidopsis* 35 MIP encoding genes comprised 13 PIPs, 10 TIPs, 9 NIPs, and 3 SIPs, and rice 34 MIPs included 12 PIPs, 10 TIPs, 10 NIPs, and 2 SIPs. For the evolutionary study of multiple genes and to support the phylogenetic tree, we performed the MIP genes’ exon–intron structural and motif analyses and ensured relevancy. Gene structure analysis revealed variations in exon from 1 to 5, while the number of introns varied from 1 to 4. There were two CAMIPs lacking introns, CATIP1-11 and CATIP1-10 (Figure 3). It has been reported earlier that an increase or decrease in the number of exon–intron can be caused by the integration and realignment of gene fragments. To gain further insight into MIP genes in pepper, CAMIPs were analyzed for conserved motifs. There were 10 conserved motifs identified; these conserved motifs distribution verified the phylogeny of the MIPs of pepper (Figure 4), and CaMIP motifs were sorted correspondingly with each CATIP, CANIP, CAPIP, and CASIP subclass. Thus, the analogous domain structure endorsed the relationships and phylogenetic classification of pepper MIPs. Furthermore, the MIPs’ motif structures were also identified (Figure 5).

### 2.4. Synteny Analysis of MIP Genes in Pepper

According to Holub’s description, it can be stated that a chromosomal territory within 200 kb containing two or more genes is called a tandem duplication event [24] (Figure 6). Seventeen pepper MIP encoding genes (*CATIP1-1*, *CAPTIP2-2*, *CAPTIP1-2*, *CANIP3-1*, *CATIP1-5*, *CAPIP1-2*, *CANIP5-1*, *CANIP1-1*, *CANIP2-1*, *CANIP6-1*, *CATIP2-3*, *CATIP3-2*, *CANIP2-2*, *CAPIP2-3*, *CAPIP2-4*, *CATIP1-12*, *CAPIP2-6*) were clustered into eight tandem duplication event locations on pepper chromosomes 01, 02, 03, 06, 09, and 10. Chr. 01 contained three clusters, indicating a hot spot of MIP gene distribution.

To further deduce the phylogenetic relationships of the pepper MIP gene family, we created comparative syntenic maps of pepper and *Arabidopsis*. CAMIPs were observed to be distributed over all five chromosomes of *Arabidopsis*. Arabidopsis MIP encoding genes were unevenly distributed in pepper MIPs on chromosomes 01, 03, 05, 06, 08, 09, 10, and 12. Among them, chromosomes 2, 3, and 4 showed that most of the MIPs were located there and were showing a linkage with chromosomes 05, 06, 08, 09, and 10 of pepper. Among *Arabidopsis* and pepper, mostly *CANIPs*, *CATIPs*, and *CAPIPs* were observed to be evolved; there were no CASIPs noticed (Figure 6).

The Ka/Ks ratio is vital in evolutionary studies in molecular biology. Here, non-synonymous substitutions (Ka) were compared with synonymous substitutions (Ks) in a gene or gene region. Ka describes the altered amino-acid sequence of the protein, while Ks are those which do not change the protein sequence. While Ka/Ks of more than 1 suggest positive selection, a Ka/Ks of less than 1 suggest purifying or negative selection. Most MIP genes were under strong purifying selection, indicating that MIP encoding genes are evolving and that is important for pepper plant survival in different environments and adaptability (Appendix A).

### 2.5. Effect of Heavy Metals on Chlorophyll, Nitrogen Content and Root Length in Pepper

Heavy metals affect plant growth in different ways. We applied As(III) at different concentrations to the pepper plant to observe the regulation of MIPs (Appendix A). As(III) at 0.5 and 1 mM was applied for 48 h and data were recorded for chlorophyll, nitrogen content, and root length. The chlorophyll content of Ck was 28.9 (nmol/cm^2^) and after treatment time was 29.5. While As(III) at 0.5 mM significantly decreased the chlorophyll content from 22.5 to 7.5, it was observed that at 1 mM As(III) the chlorophyll content decreased from 31.1 to 12.3 (nmol/cm^2^). The possible reason for this may be the high amount of ethylene produced that promotes senescence and triggers chlorophyll degradation and the disruption of the photosynthesis process.

Leaf nitrogen content was also affected by As(III). For pepper plants (Ck), leaf N content increased from 10.8 to 11.2. As(III) at 0.5 mM caused a significant decrease in N content from 9.7 to 5.2, while a decrease from 12.5 to 8 at 1 mM As(III) treatment was observed (Figure 7).

Root length is an important parameter in understanding plants’ stress response in case of heavy metal contamination in water or soil. As compared to the average root length of plants (Ck) (16.23 and 18.31 cm), respectively, for As(III) 0.5 and 1 mM, As(III) reduced average root length very significantly. As(III) at 0.5 mM concentration had an average root length of 14.1 cm, while at 1 mM it had a root length of 12.1 cm (Figure 7).

### 2.6. Tissue Differential Expression Patterns of CaMIPs

Expression patterns of CaMIPs in different tissues were carefully observed based on RNA-seq data obtained from the pepper genome sequence [25]. The generated heatmap showed a differential transcriptional abundance of the 48 CaMIPs in five major tissues, namely placenta, pericarp, leaf, stem, and root, at different developmental stages (Figure 8).

A dynamic expression was observed for CAMIPs in pepper. Some of the MIPs were highly expressed in roots (CaTIP2-2, CaPIP2-2, CaPIP2-1, CaNIP3-1, CaPIP1-6, CaTIP1-12, CaPIP2-3, CaTIP1-7, and so others), while some of the proteins were not expressed in roots, such as CaPIP1-8, CaPIP2-7, CaPIP1-4, etc. (Figure 8). CaPIP1-2, CaNIP6-1, CaTIP1-10, and CaNIP5-1 were highly expressed in stem tissue, while some proteins displayed moderate expression, though most showed low expression in stem tissues. In the leaves, only CaTIP3-1 displayed significantly high expression, while the rest of CaMIPs did not express significantly in the leaves. At the pericarp breaker stage (PC-B), only CaTIP1-13 and CaTIP1-8 exhibited high expression, while other proteins were not expressed. At the post-breaker stage (PC-B5) of the pericarp, only CaTIP1-1 was expressed, while at the post-breaker stage (PL-B10) of the placenta, CaNIP4-2 was significantly expressed, CaPIP1-1, CaNIP1-2, and CaTIP1-4 were moderately expressed, while other CaMIPs were not expressed. CaMIPs were observed at different growth stages of the pericarp and placenta at the post-anthesis stage. They were monitored for pericarp and placenta 6, 16, and 25 days post anthesis (DPA). At the placenta, post-anthesis CaTIP4-1, CaPIP1-4, and CaNIP4-3 were expressed (6DPA); at 16 DPA, CaNIP1-1 and CaTIP1-13 were significantly expressed; while at post anthesis (25DPA), two proteins, CaTIP3-2 and CaTIP4-1, exhibited high expression. During the developmental stages of the pericarp (PC), CaPIP1-7, CaPIP2-4, CaPIP1-4, CaNIP4-3, and CaPIP2-5 were significantly expressed at 6DPA. At PC-16DPA and PC-25DPA, only CaTIP1-3 was expressed at both developmental stages. At the placental mature green (PL-MG) stage, only CaTIP1-9 was highly expressed, while at the pericarp mature green (PC-MG) stage, CaTIP2-3 was significantly expressed. Two proteins, CaPIP1-8 and CaPIP2-7, did not show expression in any observed tissues (Figure 8).

### 2.7. Response of Genes Encoding Pepper MIPs to As(III) Exposure

The expression profiles of pepper MIP encoding genes were assessed in response to arsenite. As(III) at 0.5 and 1 mM (as NaAsO_2_) was applied to pepper plants, keeping plants without As(III) as Ck. Root and shoot samples were taken after 12 h and 24 h for RNA extraction followed by qRT-PCR (Figure 9).

When compared to the Ck, in the roots, only CAPIP-5, CAPIP-1-7, CAPIP1-8, and CANIP5-1 were up-regulated with As(III)0.5 at 12 h and 24 h, respectively, while the rest of the proteins were down-regulated. Up-regulated proteins were down-regulated when increasing the exposure of plants to arsenite. Some CATIPs have been shown to transport arsenite into the vacuole for sequestration. Under arsenic stress, the pepper plant might activate defensive mechanisms as a measure for the detoxification and compartmentalization of toxic arsenite and to reduce cellular damage. When As(III) 1 mM was applied to the pepper plant, CAPIP1-3, CATIP1-4, CATIP1-9, and CAPIP2-3 were shown to be up-regulated in the shoot under 12 h and 24 h stress. We observed that these up-regulated genes were down-regulated under 24 h stress conditions (Figure 9).

The prime role of most aquaporins is to transport water across cell membranes once osmotic gradients have developed by active solute transport. The water transport capability of the aquaporin monomers is low, up to 10,000 per square micron; therefore, to increase water permeability in stressed conditions, the plant has to rely on aquaglyceroporins. Aquaglyceroporins also transport glycerol. The pore diameter of the aquaglyceroporins is slightly greater than that of the water-selective aquaporins, and the pore is lined by relatively hydrophobic residues compared to the pore of a water-selective aquaporin.

In roots under As(III) 0.5 stress, CATIP1-6 CATIP1-8, CATIP1-11, CAPIP2-2 were up-regulated as compared to the control. However, these up-regulated genes displayed reduced expression at 24 h As(III)0.5 mM stress. Under As(III)1 mM stress, CANIP2-2, CAPIP2-2, CANIP1-1, and CATIP4-1 were up-regulated initially under 12 h stress, while they were observed to be down-regulated at 24 h AS(III)1 mM stress. Only CATIP1-12 was up-regulated at 24 h stress. Up-regulated CATIPs may increase the transport of arsenite into vacuoles, partitioning it from the cytoplasm to prevent damage to cellular components. Most of the down-regulated proteins of the CaMIPs family show the pepper plant response to mitigate the toxicity of arsenite exposure and to maintain normal cellular functions (Figure 9).

## 3. Discussion

China is among the leading pepper-producing countries (http://faostat.fao.org/, 2018), and the north and northwest of China are the key pepper-producing regions. Water, quite distinctly, is the universal solvent paramount for all living cells. Plants have specialized system for water and mineral transport across membranes. Among these, major intrinsic proteins are responsible for water and mineral transport, and some transport glycerol. Aquaporins and aquaglyceroporins are two major subfamilies of MIPs. Currently, aquaporin is known as being the most abundant trans-membrane transport channel of glycerol, urea, ammonia, carbon dioxide, and some metalloids other than water [3]. Major intrinsic proteins (MIPs) of plants are further divided into seven subfamilies: plasma membrane intrinsic proteins (PIPs), hybrid intrinsic proteins (HIPs), tonoplast intrinsic proteins (TIPs), small basic intrinsic proteins (SIPs), nodulin 26-like intrinsic proteins (NIPs), GlpF-like intrinsic proteins (GIPs), and uncharacterized X intrinsic proteins (XIPs) [26,27].

### 3.1. Identification, Classification, and Synteny of MIP Genes in Pepper

Pepper genome was shown to contain 48 MIP encoding genes identified after a genome-wide search that was carried out using BLAST searches. These 48 MIP encoding genes were further divided into TIPs, PIPs, NIPs, and SIP (Figure 2). Similar arrangement of MIPs have been reported in other important plant species. There were 35 MIP encoding genes identified in Arabidopsis [28], 33 in rice [29], 31 in maize [30], 34 in sweet orange [31], 60 in *Brassica rapa* [32], 55 in Populus [33], 47 in tomato [34], 71 MIPs in cotton [35], and 41 in sorghum [36], while Banana had 47 and *B. Napus* had 121 MIPs [27]. The 37 castor bean aquaporins’ genes were assigned into five subfamilies: 10 PIPs, 9 TIPs, 8 NIPs, 6 X-intrinsic proteins (XIPs), and 4 SIPs [37]. Together, these studies across diverse plant species shed light on presence of MIPs in plant biology. A chromosomal spanning region within 200 kb with two or more genes is known as a tandem duplication event [24]. In the pepper genome, we found seventeen tandem duplication genes involving TIPs, NIPs, and PIPs on eight chromosomes (Figure 6). The apprehension of the frequency of chromosomal duplication inter and intra species exhibits a species-specific and shared evolution process. During the evolution process, the plant genomes have been reshaped and restructured by different segmental duplication events like translocation, inversion, and whole genome duplications [38].

### 3.2. Phylogeny, Characteristics, and Chromosomal Localization of MIP Genes

The phylogenetic association of *Arabidopsis thaliana*, *Oryza sativa*, and *Capsicum annum* were elucidated (Figure 2). Pepper’s MIP encoding genes were divided into CaPIPs, CaTIPs, CaNIPs, and CaSIPs. Amino-acid numbers varied from 107 (CATIP2-1) to 329 (CANIP5-1) for MIP genes in pepper, presenting MIP gene diversity and variation in the pepper genome (Table 1). Variation is exhibited across different plant species related to A.A in MIPs, while there were 75–349 A.A for genes encoding MIPs on the sweet orange genome [31]. The pH at which a molecule carries no net electrical charge or is electrically neutral is known as the isoelectric point (Ip). Most of the MIPs exhibited high Ip. Different plant species observed high Ip for MIPs, for example, castor bean, cucumber, and chickpea [37,39,40]. On the basis of identification and prediction of sub-cellular localization of MIPs, it was observed that MIPs were distributed into different groups on the basis of their phylogeny (Table 1). Almost all CATIPs were localized in the vacuole, CAPIPs were observed in the plasma membrane, while CANIPs were localized in nodules. We did not find any pepper MIPs localized in mitochondria, cytoplasm, or the nucleus, while in chickpea, TIPs were localized in mitochondria and cytoplasm [40]. MIP encoding genes were dispersed on 11 chromosomes out of 12 pepper chromosomes (Figure 1). Chromosomes no. 1 and 6 contained the highest number of genes encoding MIPs, while CsMIPs in *B. napus* and orange were observed to be distributed over all chromosomes [27,31]. The instability index is the measurement of stability of any protein in a test tube; here, all of the CAMIPs were stable. When the instability index < 40, the proteins are said to be stable [41].

### 3.3. Spatio-Temporal Expression of CAMIP Encoding Genes

The tissue-specific expression of MIPs encoding genes are closely linked to the physiological functions of certain tissues. We examined the expression profiles of pepper MIPs (Figure 8). We observed distinct and contrasting transcript levels of MIPs among the different tissues. Pepper TIPs, SIPs, and NIPs were expressed in roots as compared with Arabidopsis, maize, potato, and tomato, where MIPs were significantly expressed in major tissues, including roots. Earlier, some TIPs, PIPs, NIPs, and SIPs in tomato were highly expressed in leaves, root, flower, 3 DAP, 7 DAP and breaker stage [34]. In our study, specific NIP genes in pepper, such as CANIP4-1 and CANIP3-1, were also preferentially expressed in roots, supporting the notion that NIPs are functionally associated with root-related physiological roles. A similar predominant expression of wheat NIPs (TaNIP1-7, TaNIP4-1C1, TaNIP1-1C1, TaNIP1-1C2 and TaNIP2-1C1) in root tissues confirmed NIPs’ role in roots [42]. The pepper genome appears to contain only a single CAPIP gene, which showed no detectable expression in the tissues we analyzed, while all wheat PIPs were expressed across various tissues [34]. From these results, it can be assumed that PIP expression may be restricted to specific developmental stages, or some may be stimulus dependent. Pepper SIPs, TIPs and NIPs were not expressed prominently in fruit tissues as reported earlier in Banana [43]. Furthermore, rice MIPs (PIPs, NIPs, and TIPs) exhibited organ specificity in the rice genotypes studied [29]. Collectively, pepper MIPs exhibited dynamic and tissue-specific expression, underlining the vitality of further functional studies to elucidate the regulatory mechanisms and biological and physiological roles of the pepper MIPs.

### 3.4. Effect of Arsenite on the Physio-Morphology in Pepper

Soil contamination with arsenic and the chemically similar metalloid antimony has emerged as an environmental concern threatening the health of the environment, plants and humans [44]. Arsenic contamination in water and soil was shown to disrupt both physiological and biochemical processes in plants, resulting in decreased plant productivity [45]. Upon increased exposure to arsenic, plant roots responded by a variety of physiological and biochemical changes, and to survive, plants must activate various responses such as efflux and complexation to mitigate the toxic effects of arsenic [46]. In our experiment, As(III) significantly impaired the pepper plant thereby reducing leaf chlorophyll content at higher concentrations when compared with Ck (Figure 7). These findings align with previous observations, where As(III) exposure led to reduced photosynthetic leaf pigments, the degradation of the chloroplast membrane, as well as a decreased enzymatic activities by reacting with the sulfhydryl group of proteins [47]. Additionally, arsenic exposure disrupted nutrient balance as well as protein metabolism, which eventually caused chlorophyll loss and impaired photosynthesis [46]. Antimony, frequently found alongside arsenic due to its similar chemical properties and sharing biogeochemical cycles, has been shown to have negative effect on plant health [48]. High concentrations of Sb generated reactive oxygen species (ROS), inducing oxidative stress and effecting photosynthesis and carotenoid synthesis [44,49,50]. We also observed a significant decline in leaf nitrogen content (N content) in response to As(III) exposure (Figure 7). This could be attributed to chlorophyll degradation, impairing the photosynthetic machinery and reducing photosynthetic efficiency. Nitrogen assimilation is an energy-demanding process that will be disrupted by the impaired photosynthesis, thereby limiting the energy that is necessary for nitrogen metabolism [45,51]. Furthermore, pepper plant root length was significantly reduced in response to higher concentrations of As(III). It seems likely that this inhibition is a result of the arsenic-induced disruption of cell elongation and mitosis processes in root meristematic cells, leading to decreased cell proliferation. Our current findings were in alignment with earlier research demonstrating the inhibitory effects of Sb on root development in various plants, including rice sprouts [49,50,52]. In light of these findings, there is an increased interest in understanding the signal transduction pathways participating in plant responses to arsenic and antimony stress. ROS, ethylene, and auxin related pathways are believed to play key roles in mediating plant defense mechanisms. Exploring these pathways may create insights into potential strategies for enhancing plant tolerance to metalloid toxicity.

### 3.5. Response of CAMIPs to Arsenite Stress

Arsenic exists in soil and water environments in both inorganic (arsenite, As(III); arsenate, As(V)) and organic (methylated) forms, while inorganic As species are the predominant forms and thus more toxic to plants [53]. Major intrinsic proteins (MIPs), including aquaporins and aquaglyceroporins, are membrane-associated channel proteins facilitating the bidirectional, concentration-dependent transport of water, small solutes, and some heavy metals, including As(III), across membranes. Upon exposure to As(III) and Sb(III), plants can and will respond by increased efflux out of the cell or into the tonoplast. However, without reducing the uptake of As(III) and Sb(III), the energy requiring efflux would constitute a futile cycle. Therefore, plants must regulate and thereby decrease uptake of both As(III) and Sb(III). The situation is quite different for environmental exposure to arsenate As(V) and antimonate Sb(V). Here, the uptake of As(V) and subsequent reduction to As(III) would result in a concentration gradient of high internal As(III) and low external As(III), leading to concentration-dependent efflux by a facilitator of the MIP family. This MIP-facilitated efflux does not require energy for transport. It is important to remember that not all members of the MIP family are involved in the transport of the metalloids As(III) and Sb(III). MIPs encompass several subfamilies such as nodulin 26-like intrinsic proteins (NIPs), tonoplast intrinsic proteins (TIPs), plasma membrane intrinsic proteins (PIPs), hybrid intrinsic proteins (HIPs), GlpF-like intrinsic proteins (GIPs), and X intrinsic proteins (XIPs) [1]. Among these, NIPs have been extensively reported to mediate As(III) transport. In pepper plants treated with As(III) for 24 h, the shoots’ MIPs facilitated As(III) transport at both lower and higher concentrations (Figure 9). Some CANIPs and CATIPs in the shoots were up-regulated in response to As(III). Our findings are in line with previously reported nodulin 26-like intrinsic proteins (NIPs), which were involved in As(III) transport in plants [51,54,55]. For instance, the rice *Lsi1* (OsNIP2;1) mutant showed reduced silicon and As(III) uptake, and accumulation, confirming NIPs’ role in root uptake and xylem loading [55]. In pepper, short-term As(III) exposure (12–24 h) led to dynamic and concentration-specific MIPs expression in shoots and roots. Importantly, CANIPs and CATIPs were up-regulated in shoots, whereas there was a variable response in CAPIPs. Some CAMIPs were down-regulated, likely as a detoxification execution in pepper. In roots, MIPs such as CATIP1-9 and CATIP1-2 were up-regulated at high As(III) levels, while others showed reduced expression, indicating selective arsenic transport and stress adaptation. Beyond rice and pepper, *OsNIP1;1*, *OsNIP2;2*, *OsNIP3;1*, *OsNIP3;2*, and *OsNIP3;3*, were significantly involved in arsenic transport in yeast assays [56,57]. In *Xenopus oocytes*, *OsPIP2;4*, *OsPIP2;6*, and *OsPIP2;7* also transported As(III), suggesting some PIPs may assist As(III) influx and efflux [58]. When *PvTIP4;1* was over-expressed in *A. thaliana* roots, it enhanced As(III) accumulation and sensitivity, although the functional roles of TIPs remain unclear [59]. The MIPs’ selectivity extends beyond arsenite and water to the gases (e.g., NH₃, CO₂) and ions (K⁺, Cl⁻) [60], implying panoramic physiological relevancy. Until now, only a fraction of plant MIPs have been functionally characterized. The regulatory mechanics, including transcriptional and post-translational modifications (e.g., *MAPK*, calcium-dependent kinases) as well as the phyto-hormonal signaling (e.g., jasmonic acid, ethylene, and ABA) influence the MIPs’ activity during arsenic stress. Transcriptomic studies revealed that root-related MIPs in rice were down-regulated under As(III), while some shoot-expressed OsNIPs and OsPIPs were up-regulated, indicating tissue-specific control of arsenic transport. In response to As(III), genes encoding MIPs showed dynamic and specific expression; at low As(III) concentration with 24 h exposure, mostly CATIP proteins were up-regulated (Figure 9). Previously, TIP and PIP, both being deferentially expressed in response to As(III), were tested as potential As(III) transporters in rice, [51], Arabidopsis [61], and rice roots [62]. In response to higher concentrations of As(III), CATIP1-9 and CATIP1-2 were up-regulated at 12 h exposure, while the remaining MIPs were not significantly affected. Earlier findings suggested that the PIPs and TIPs that facilitated osmotic water transport across permeable membranes might be source of increasing As(III) concentration in root cells [29,62]. It was also observed that OsNIP2;1 is localized on the distal side of the plasma membrane and shown to be responsible for As(III) uptake from soil into the root cells [55,63]. Investigations showed that short-term As(III) exposure for four hours resulted in both active influx and efflux of As in roots, suggesting a bidirectional As(III) permeability of PIPs in rice plants. So, it cannot be confidently concluded that MIPs are up-regulated or down-regulated in response to As(III) only. It has been made possible with extensive bioinformatics studies and biochemical and molecular studies to determine that various subfamilies of MIPs (aquaporins and aquaglyceroporins) have a diversity of substrate specificities, subcellular localization, and modes of regulation of these MIPs.

The regulation of CAMIPs under As(III) stress is critical for maintaining arsenic homeostasis by reducing the uptake of As(III) in plants. In yeast, the expression of *Fps1* was reduced within 15 min of As(III) exposure, but increased 3-fold after 4 h, indicating dynamic regulatory mechanisms [64]. In plants, MIPs are strictly regulated in response to fluctuating mineral availability and are expressed variably under limited or surplus minerals to avoid toxicity [65]. Transcriptome analysis in rice revealed that 17 out of 20 MIPs were down-regulated in roots (e.g., *OsNIP1;1*, *OsNIP2;1*, *OsNIP2;2*, *OsPIP2;6*, *OsPIP2;4*, *OsNIP3;1)*, while *OsNIP1;1*, *OsNIP2;2*, *OsNIP3;1*, and *OsPIP2;7* were up-regulated in shoots in response to As(III) stress [51], indicating the tissue-specific transition of arsenic transport control within plant. *OsARM1* (Arsenite-Responsive MYB 1) was recently identified as a negative regulator of As(III) in rice [66]. Furthermore, several miRNAs have been implicated in controlling arsenic signaling and transport by targeting genes involved in transcriptional regulation, nutrient transport and signal transduction, metabolism, and development [67,68]. For instance, mitogen-activated protein kinases (*MAPKs*) regulate yeast MIPs, while Ca^2+^-dependent protein kinase *AtCPK31* in *Arabidopsis thaliana* positively regulates As(III) uptake through *AtNIP1;1* [69], and calcium channels and NADPH oxidases are activated, triggering calcium signaling cascades upon arsenic entry into the cell [63].

## 4. Materials and Methods

### 4.1. Identification of MIP Homologous Sequences

MIP conserved domains based on a Hidden Markov Model (HMM) (PF00230) were obtained from the Pfam database (http://pfam.sanger.ac.uk/, accessed on 1 September 2024). The amino-acid sequence of the MIP domain was employed to search for possible MIP domain homolog hits on the entire genome sequence of the MIP sequence of pepper, rice, and Arabidopsis, from (http://peppergenome.snu.ac.kr/) for pepper, (http://www.phytozome.net/rice.php) for rice OsMIP sequences, and Phytozome (http://www.phytozome.net/arabidopsis.php) for Arabidopsis AtMIP sequences, to MIP encoding gene sequences, respectively, accessed on 1 September 2024 [70]. To identify the genes encoding MIP of *Capsicum annum* L., we used the HHM profile of the MIPs domain as a query to execute a HMMER search (http://hmmer.janelia.org/) in the entire pepper genome. All non-redundant sequences encoding complete MIP domains were entirely screened out to be putative genes encoding MIP. Each non-redundant sequences of genes encoding MIP was double restrained for the presence of the conserved MIPs domain by the utilization of SMART search (http://smart.emblheidelberg.de/, accessed on 1 September 2024).

### 4.2. Characteristics and Chromosomal Localization of MIP Genes

The individual genes encoding MIPs were examined regarding their length, no. of amino acids (A.A), and no. of open reading frames (ORF) using the MIP gene’s complete CDSs. ATG was set as the start codon using ORF finder (https://www.ncbi.nlm.nih.gov/orffinder/). The molecular weights and isoionic point (IP) of the genes encoding the MIP family for pepper were computed using the ExPASy server (http://web.expasy.org/compute_pi/) [71,72]. In addition, the instability index and aliphatic index were calculated using the ExPASy server (https://web.expasy.org/cgi-bin/protparam/protparam). The start and stop points were calculated from the pepper genome for genes encoding MIP, and Mapinspect was employed to locate 48 genes encoding MIP on 12 chromosomes according to the first nucleotide as the start site of the gene. Characteristics of MIPs were calculated using the above mentioned websites accessed on 1 November 2024.

### 4.3. Phylogeny and Sequence Similarity of Pineapple MIP Homologs

We used amino-acid sequences for multiple sequence alignment for the MIPs from Arabidopsis, rice, and pepper, employing MUSCLE (v5.0-5.3) with default settings. Subsequently, MEGA 6.0 software was used to make an unrooted phylogenetic tree based on the Neighbour-Joining (NJ) procedure with the following traits: JTT model, pairwise gap deletion, and 1000 bootstraps [73]. Furthermore, maximum likelihood, minimal evolution, and PhyML methods were also practiced for the tree construction to validate the results of the NJ method.

### 4.4. Gene Structure Analysis and Conserved Motif Identification

The structural diversity and evolutionary aspect of the genes encoding MIP from pepper were explored by studying the gene structure (exon–intron organization). The exon–intron positions of the genes were ascertained using the Gene Structure Display Server (http://gsds.cbi.pku.edu.cn) through a comparison of their full-length coding sequence (CDS) [74] with their corresponding genomic DNA sequences from Phytozome (http://www.phytozome.net/pepper.php, accessed on 1 October 2024) [74], while the amino-acid sequences of the 48 MIPs were analyzed and statistically identified by MEME (Multiple EM for Motif Elicitation) (http://meme-suite.org/tools/meme) with the motif length set to 6–100 and motif sites to 2–120, website were accessed on October 2024. The maximum 10 motifs were identified; the statistical organization of one single motif was ‘any number of repetitions’, and the other parameter was ‘search given strand only’.

### 4.5. Synteny Analysis and Collinearity Analysis of MIP Genes in Pepper

Gene duplication events were identified by utilizing the Multiple Collinearity Scan toolkit (MCScanX) with the default settings accessed on October 2024 [38]. To display the synteny relationship of the orthologous genes encoding MIP acquired from pepper and Arabidopsis genome, the syntenic analysis maps were designed using the Dual Systeny Plotter software accessed on 1 October 2024 (https://github.com/CJ-Chen/TBtools). Ka (Nonsynonymous)/Ks (Synonymous) analysis was performed by the “One Step MCScanx” and “Simple Ka/Ks Calculator” functions, following the procedure described earlier [38].

### 4.6. Plant Material, Growth Conditions, and Heavy Metals Assay

The seeds of the experimental pepper (*Capsicum annum* L.) plants’ seeds (chao tian jiao variety) were provided by the Institute of Environmental Microbiology, Fujian Agriculture and Forestry University, washed with H_2_O_2_ (10%) for 10 min, then rinsed with distilled water, and then the cleaned seeds were germinated on moist tissue on a Petri dish. After germination, all seedlings were placed (three replicates) on an opaque box with a 2L capacity (4 plants/bottle) filled with a regularly aerated nutrient solution [75]. The nutrient solution with pH = 5.6 ± 0.1 contained 0.505 mM KNO_3_, 0.150 mM Ca(NO_3_)_2_·4H_2_O, 0.1 mM NH_4_H_2_PO_4_, 0.1 mM MgSO_4_·7H_2_O, 4.630 mM H_3_BO_3_, 0.910 mM MnCl_2_·4H_2_O, 0.030 mM CuSO_4_·5H_2_O, 0.060 mM H_2_MoO_4_·H_2_O, 0.160 mM ZnSO_4_·7H_2_O, 1.640 mM FeSO_4_·7H_2_O, and 0.810 mM NA_2_–EDTA. The installed climate chamber was set to a daily photoperiod of 16 h at 26–27 °C with 230 µmol m^−2^ s^−1^ photon flux, followed by a night period of 8 h at 24 °C. Pepper plants were grown for 5–6 weeks, while nutrient solutions were changed twice a week. For heavy metal treatment, plants were subjected to arsenite when the plants had 6 or 8 true leaves. As(III) was applied in different concentrations as NaAsO_2_ (0.5 and 1 mM) was directly applied to nutrient solution, and sampling was carried out for RNA extraction at 12 h and 24 h intervals. Plants before the application of As(III) were considered as Ck.

### 4.7. Leaf Chlorophyll, Nitrogen Content, and Root Length

Chlorophyll and nitrogen content was determined at the early vegetative growth stage (10 leaves stage) of pepper using the SPAD 502 (Minolta Camera, Co., Osaka, Japan) chlorophyll meter; observations were performed on fresh, fully opened leaves from 3 to 5 randomly selected plants in each treatment after 48 h. Five chlorophyll meter readings were taken around the midpoint of each leaf avoiding the midrib for each plant [76]. Three plants were sampled for chlorophyll measurement, while the root length of treated plants was measured using a ruler. Root length was taken from the intersection of the stem and root starting point. The roots of three plants were measured, the mean was taken, and three plants were sampled for root length.

### 4.8. RNA Extraction and qRT PCR

The transcriptional responses to As (III) treatments of genes encoding MIPs were assessed by qRT-PCR. First, six leaves, and two plants’ roots were sampled after the application of AS(III) at 0.5 and 1 mM at 12 h and 24 h intervals (Ck—without application of As(III)), stored in liquid nitrogen, and then moved to −80 °C storage. There were three biological repeats harvested and analyzed for each treatment. Each sample was comprised four individual plants. A bioline kit (ISOLATE II RNA Plant Kit, Meridian Bioscience, Beijing, China) was used following manufacturer protocol for RNA extraction. cDNA synthesis was performed as follows: 1 µg of total RNA was obtained with Bio-Rad iScript cDNA synthesis kit (Bio-Rad Laboratories (Singapore). For qRT-PCR, a mixture containing 12.5 µL of SYBR Green (Bioline), 1 µL (10 µM) of forward and reverse primer, 5.5 µL RNase free water, and 5 µL cDNA was used [77]. The data standardization was performed using a reference gene, CaActin. The qPCR cycle conditions were as follows: 95 °C for 3 min, 40 cycles of 95 °C for 15 s, followed by 60 °C for 45 s. Melt curves for each gene were noted at the end of each cycle. The primers used in qPCR are given in Appendix A. The relative gene expression was calculated using the geometric mean of Ct (threshold cycles) values [78] from the reference gene CaActin employing the 2^–ΔΔCt^ method [79]. The expression profiles obtained from qRT-PCR were presented in a heatmap [80], while the tissue-specific expression profiles of CAMIPs were assessed using publicly available RNA-Seq [25].

### 4.9. Statistical Analysis

All results are expressed as the means ± standard error (SE) from at least three repeats in the experiment. To analyze statistical significance, a two-tailed Student’s *t*-test was used.

## 5. Conclusions

To our knowledge, this is the first genome-wide study of genes encoding members of the MIP family in pepper. Precise phylogeny assigned the 48 MIP encoding genes to five subfamilies based on the sequence similarity as CANIPs, CATIPs, CAPIPs, and CASIPs according to their relationship with Arabidopsis and rice counterparts. Furthermore, the structural and physio-chemical properties of the encoded CAMIPs were investigated using bioinformatics tools followed by an assessment of the CAMIPs’ expression profiles with publicly available deep transcriptome sequencing. Several CAMIPS exhibited tissue-specific expression, with some CANIPs expressed in root tissues. Morphological features were observed, and leaf chlorophyll and nitrogen contents were observed after exposing pepper plants to As(III), where it was shown that As(III) significantly reduced pepper plant leaf chlorophyll and leaf nitrogen content, and reduced root length at higher concentrations as compared to the control. The transcript levels of pepper MIP encoding genes after exposure to As(III) were also obtained through qRT-PCR. Facilitated transport of As(III) was observed in both shoot and root samples treated with a high concentration of As(III). Some of the genes encoding MIPs were up-regulated in response to higher concentrations of As(III) in roots and shoots and then down-regulated as the As(III) stress was extended. CANIP2-2, CAPIP2-2, CANIP1-1, and CATIP4-1 can be important for future research for their active role in arsenite transport in roots and shoots. Our study provides a basis for the further use of molecular approaches to obtain a predictive functional study of genes encoding members of the MIP family and to bio-engineer pepper plants with reduced arsenic uptake in natural environments.

## Figures and Tables

**Figure 1 plants-14-01475-f001:**
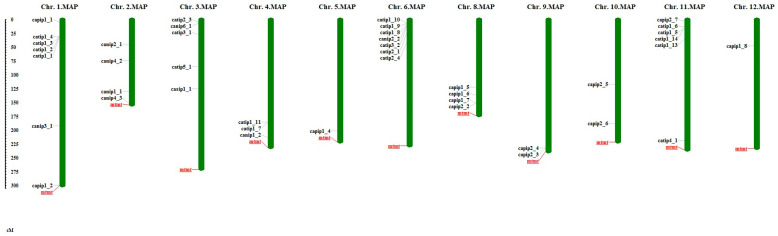
The genomic distribution of MIP genes on pepper chromosomes. Chromosome size is indicated by mtmt in Mbs, while genes are located according to their start point on respective chromosomes. The pattern bar graph shows the MIP genes located on each chromosome. The *Mapinspect* program was used to locate CAMIP genes on chromosomes.

**Figure 2 plants-14-01475-f002:**
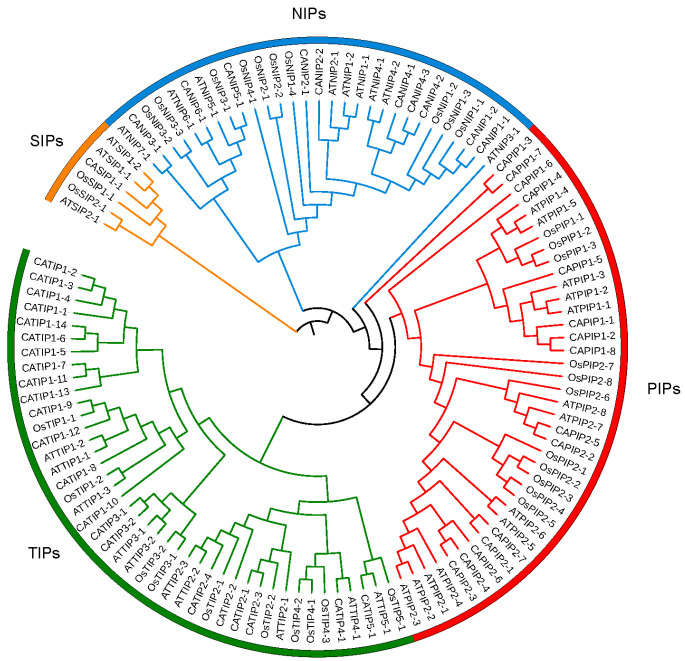
Phylogenetic analyses of MIPs from pepper, rice, and *Arabidopsis*. The phylogenetic tree was constructed using the neighbor-joining (NJ) method with the following parameters: JTT model, pairwise gap deletion, and 1000 bootstraps. *MEGA6.0* was used to make the phylogenetic tree. Subfamilies were indicated by different colored lines: SIPs (orange); NIPS (blue); PIPs (red); and TIPs (green).

**Figure 3 plants-14-01475-f003:**
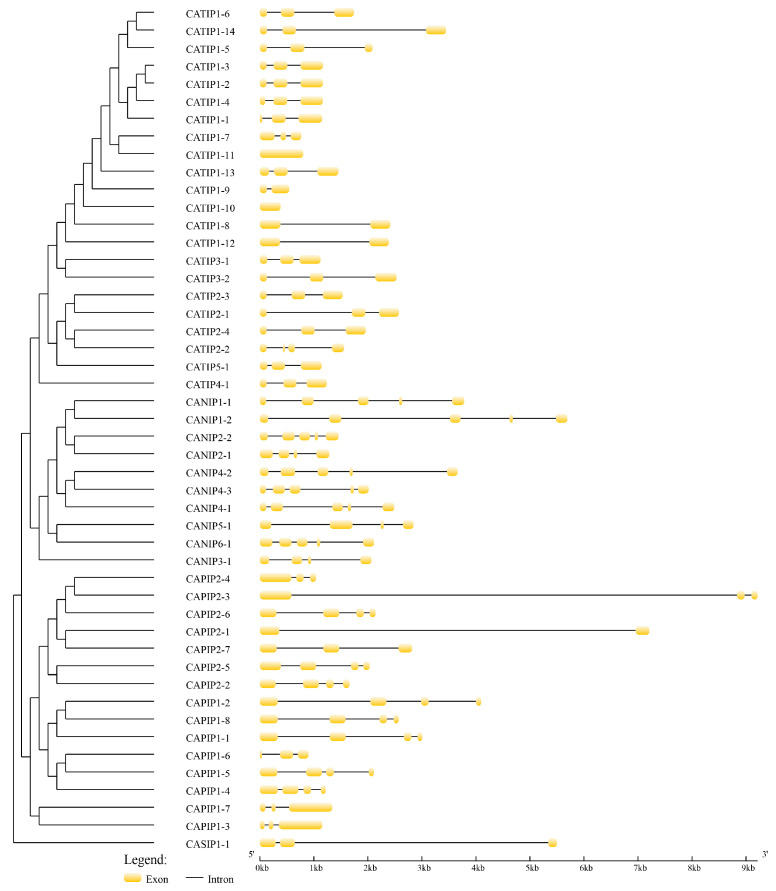
Exon–intron structure of the 48 pepper MIP genes. Yellow regions indicating exons and introns are shown as black lines. Gene structures of all CAMIP encoding genes were visualized using the Gene Structure Display Server (GSDS) program.

**Figure 4 plants-14-01475-f004:**
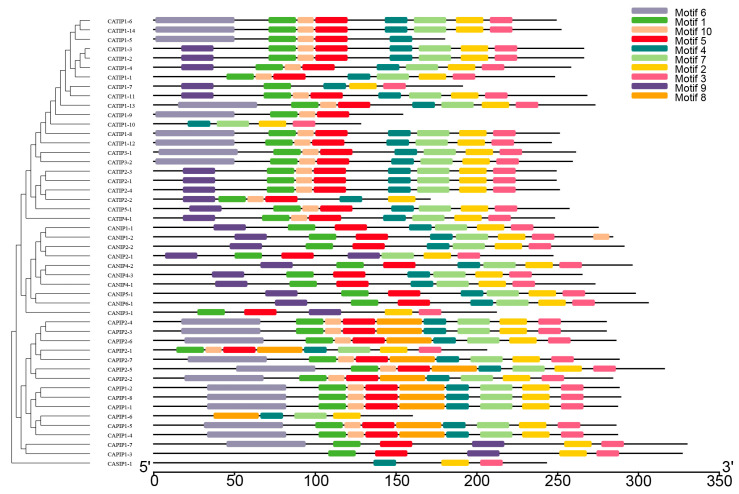
Conserved motifs of MIPs in pepper. These motifs were identified by MEME tool using protein sequences. Ten motifs (1–10) are indicated by different colored numbers.

**Figure 5 plants-14-01475-f005:**
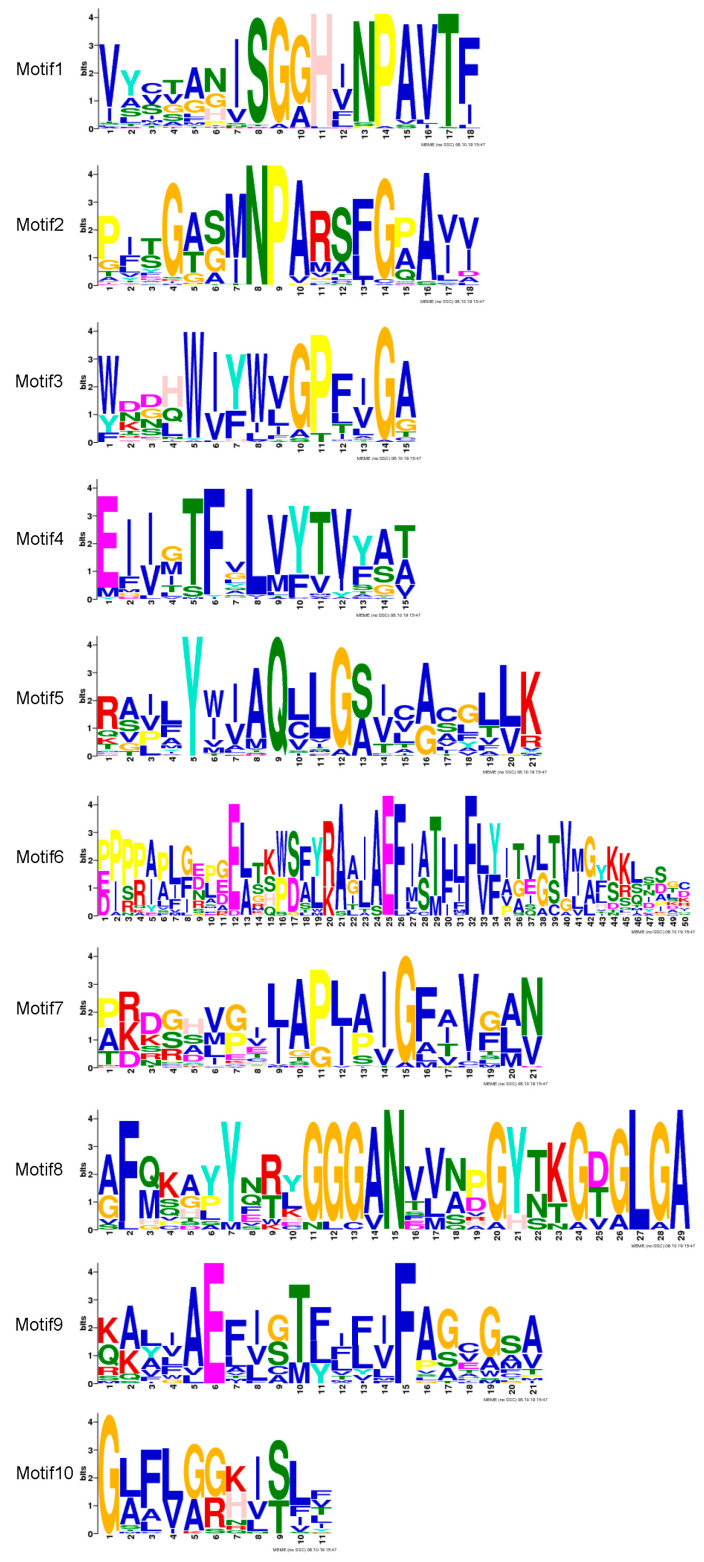
Sequence logo of CAMIPs domains obtained through the MEME program. The *x*-axis indicates the conserved sequences of these domains. The overall heights of each stack represent the degree of conservation at each position, while the heights of the letters within each stack indicates the relative frequency of amino acids.

**Figure 6 plants-14-01475-f006:**
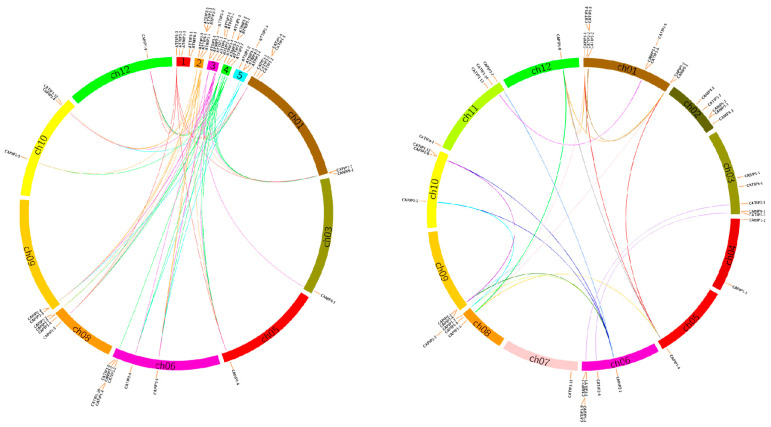
Synteny analysis of pepper MIP genes and Arabidopsis MIP genes. Different colored lines connecting two chromosomal regions indicates syntenic regions between pepper (1–12) and Arabidopsis (1–5) chromosomes. Gene duplication events were identified by utilizing Multiple Collinearity Scan toolkit X version (MCScanX). Syntenic analysis maps were designed using the Dual Systeny Plotter software TBtools (https://github.com/CJ-Chen/TBtools, accessed on 1 November 2024).

**Figure 7 plants-14-01475-f007:**
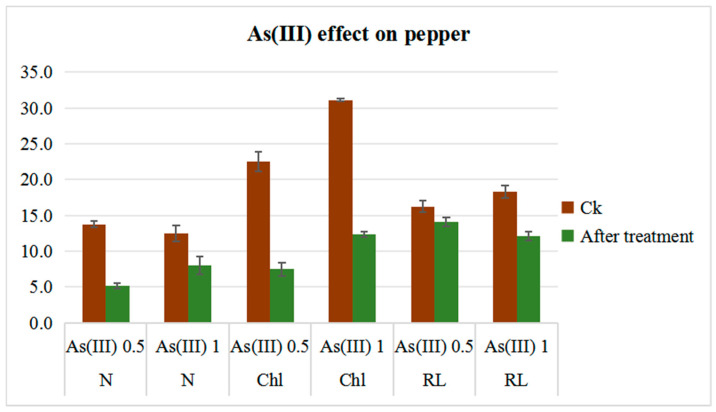
Nitrogen content (nmol/cm^2^) (N), chlorophyll content (nmol/cm^2^) (Chl), and root length (cm) (RL) measured at the 10 leaves stage at the middle leaves using SPAD 502 equipment. As(III) was applied in 0.5 and 1 mM. Error bars are presented as black lines on top of each bar. N, Chl, and RL were significantly different with control.

**Figure 8 plants-14-01475-f008:**
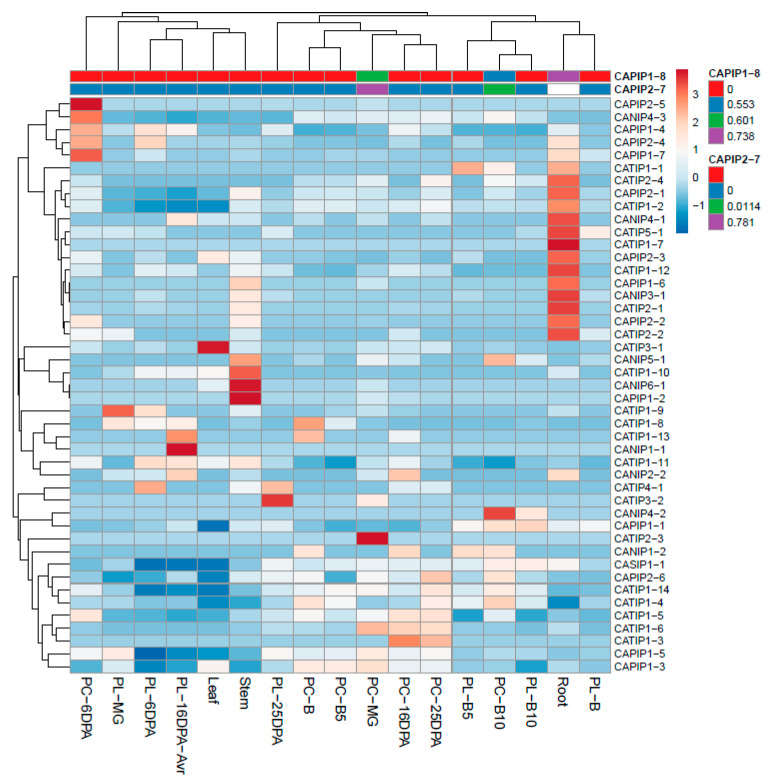
Expression profiles of the 48 genes encoding CaMIPs in leaf, stem, root, different stages of pericarp and placenta development in pepper [25]. Pericarp days to post anthesis (PC-DPA), placenta days to post anthesis (PL-DPA), pericarp mature green (PC-MG), placenta mature green (PL-MG), breaker stage (B), and the CaMIP gene expression (RPKM) scale are on the right side of the heatmap. Dark blue indicates low expression and dark brown indicates high expression.

**Figure 9 plants-14-01475-f009:**
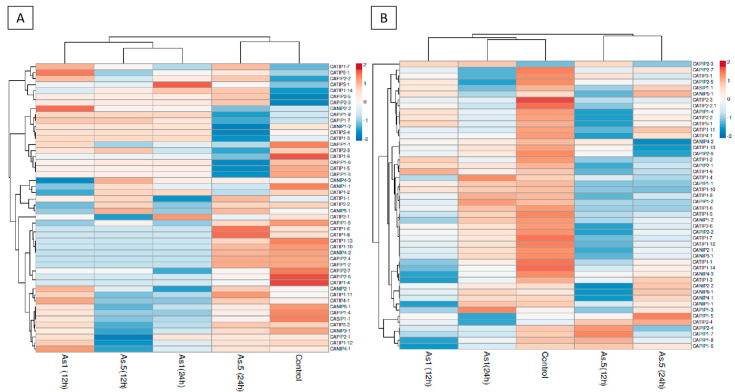
Expression profiles of genes encoding CAMIP to As(III) in pepper (qRT-PCR) shoot (**A**) and roots (**B**). Down-regulated genes are shown in blue while up-regulated genes are shown in red. As 0.5 (As(III) 0.5 mM for 12 and 24 h), As1 (As(III) 1 mM for 12 and 24 h). Each experiment contained three biological replicates (sampling n = 3).

**Table 1 plants-14-01475-t001:** Genomic and molecular characterization of pepper MIP genes.

Gene ID	Gene Names	Chr. No.	Length of A.A	Subcellular Localization	IP	No. of ORFs	MW (Da)	I.I	A.I
CA12g14150	*CAPIP1-8*	12	279	PM	8.3	6	30,858.9	32.1	100.4
CA11g19000	*CAPIP2-7*	11	286	PM	7.7	5	30,657.7	32.3	97.0
CA11g14810	*CATIP1-14*	10	287	PM, V	7.7	7	26,998.9	30.8	96.3
CA11g13570	*CATIP1-13*	10	288	V	8.7	9	29,363.2	31.7	98.7
CA11g00880	*CATIP4-1*	11	286	V	6.0	6	25,849.4	33.6	99.7
CA10g15710	*CATIP1-12*	Scaffold	283	V	6.7	6	4950.5	29.4	102.8
CA10g15210	*CAPIP2-6*	10	287	PM	8.2	8	30,543.4	28.5	105.1
CA10g07050	*CAPIP2-5*	10	250	PM	9.2	9	33,792.1	27.7	113.2
CA09g02780	*CAPIP2-4*	9	258	PM	8.5	8	29,778.7	28.2	110.5
CA09g02770	*CAPIP2-3*	9	260	PM	8.5	7	29,803.7	30.1	107.4
CA08g19380	*CAPIP2-2*	8	245	PM	8.9	10	30,286.2	19.8	112.3
CA08g18710	*CAPIP1-7*	8	248	PM	7.6	7	35,282.7	24.2	117.3
CA08g15090	*CAPIP1-6*	8	248	PM	6.1	3	16,965.4	26.2	116.1
CA08g07290	*CAPIP1-5*	8	247	PM	8.8	8	30,616.5	25.1	113.3
CA07g03360	*CATIP1-11*	4	256	PM	6.0	6	28,374.8	36.3	94.2
CA06g24330	*CATIP1-10*	6	265	V	4.2	2	13,484.5	21.9	100.6
CA06g24320	*CATIP1-9*	6	265	V	9.8	3	15,630.2	21.2	102.1
CA06g23450	*CATIP1-8*	6	283	V	6.0	9	25,663.8	29.5	102.0
CA06g22180	*CANIP2-2*	6	274	CM	9.0	8	31,634.0	26.9	100.0
CA06g20630	*CATIP3-2*	6	272	V	7.1	9	27,254.7	34.2	113.3
CA06g12470	*CATIP2-4*	6	205	V	5.7	3	25,180.3	29.2	101.4
CA06g08890	*CAPIP2-1*	Scaffold	272	PM	9.5	6	22,112.8	16.8	99.4
CA05g16220	*CAPIP1-4*	5	251	PM	9.1	6	30,731.9	30.5	115.1
CA04g14660	*CAPIP1-3*	Scaffold	257	PM	7.1	7	34,863.9	21.7	97.6
CA04g01090	*CANIP1-2*	4	264	PM	7.8	5	30,104.8	22.7	117.4
CA03g34630	*CATIP2-3*	3	267	V	6.1	8	24,908.1	32.4	106.0
CA03g32190	*CANIP6-1*	3	295	PM	8.6	5	31,366.6	36.6	111.6
CA03g23890	*CATIP3-1*	3	247	V	8.9	7	27,687.2	26.6	100.8
CA03g15670	*CATIP5-1*	3	248	PM, V	8.9	7	26,699.1	30.1	116.6
CA03g14380	*CASIP1-1*	3	305	PM, V	9.1	4	25,994.6	28.8	100.5
CA02g24190	*CANIP4-3*	2	221	PM	8.9	6	28,068.2	25.2	111.2
CA02g13230	*CANIP1-1*	2	297	PM	9.3	4	28,950.5	32.4	96.0
CA02g13220	*CANIP2-1*	2	211	PM, V	9.2	2	26,444.2	25.6	114.6
CA02g07620	*CATIP1-7*	4	127	PM, V	5.7	4	20,386.3	22.6	101.4
CA02g06180	*CANIP4-2*	2	326	PM	6.7	9	31,482.8	29.1	115.4
CA01g34840	*CANIP5-1*	Scaffold	329	PM	8.9	7	30,914.9	40.6	118.0
CA01g34180	*CAPIP1-2*	1	153	PM	8.3	6	30,766.7	29.4	107.3
CA01g24170	*CATIP1-6*	10	179	PM, V	6.3	7	26,368.8	30.6	112.8
CA01g24160	*CATIP1-5*	10	170	PM, V	7.9	2	19,540.2	29.8	108.0
CA01g24010	*CANIP3-1*	1	125	PM, V	9.3	7	21,990.9	30.8	105.4
CA01g07110	*CATIP1-4*	1	119	PM, V	5.7	6	27,566.0	30.5	102.5
CA01g07100	*CATIP1-3*	1	190	V	5.5	5	28,142.7	23.5	90.9
CA01g07090	*CATIP1-2*	1	114	PM, V	5.7	5	28,126.7	15.6	140.2
CA01g05540	*CATIP2-2*	Scaffold	242	V	6.4	4	17,422.3	32.2	109.6
CA01g02480	*CAPIP1-1*	1	159	PM	7.7	7	30,633.6	31.7	95.0
CA00g62940	*CATIP1-1*	1	124	V	6.3	7	26,388.5	31.6	92.8
CA00g57100	*CANIP4-1*	Scaffold	121	PM	8.4	7	29,060.1	24.8	82.2
CA00g46880	*CATIP2-1*	6	107	V	6.0	6	25,060.2	29.8	93.9

Note: Chr = chromosome; MW = molecular weight; IP = iso-electric point; A.A = amino acid; ORF = open reading frame; V = vacuole; PM = plasma membrane; instability index (I.I); aliphatic index (A.I). ExPASy server (http://web.expasy.org/compute_pi/, accessed on 1 November 2024) was used to quantify A.A, MW, PI. I.I, and A.I.

## Data Availability

Data are contained within the article and the Appendix A.

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
