# Peer review of "Genomic Survey of Genes Encoding Major Intrinsic Proteins (MIPs) and Their Response to Arsenite Stress in Pepper (Capsicum annum)"

_plants, 2025, doi:10.3390/plants14101475_

Round 1

Reviewer 1 Report

Comments and Suggestions for Authors

General comment

The manuscript presents an interesting study on the effects of Arsenite (AsIII) on pepper (Capsicum annum) morpho-physiology and associated gene expression responses. One of the main strengths of the work lies in its integrative approach, combining physiological observations with transcriptomic analysis to investigate plant responses to a toxic environmental contaminant. The topic is of clear interest to the plant biology and environmental stress research communities. Nevertheless, while the study has potential, the manuscript in its current form requires substantial revision to enhance clarity, logical consistency, and experimental rigor.

One of the major issues lies in the structure and content of the Introduction. The narrative currently lacks a clear logical flow, with redundant and fragmented information dispersed throughout the section. In addition, the aims and broader significance of the study are not explicitly defined, which undermines the overall rationale for the research. From a methodological perspective, the experimental design is insufficiently described. Several critical steps in the workflow are either missing or not adequately justified, making it challenging to understand the reasoning behind specific methodological choices.

While the presentation of results is generally clear, the section analyzing the impact of As(III) on plant morpho-physiology is confusing. In particular, it is unclear what constitutes the control in these experiments, whether it refers to the untreated plants sampled concurrently with treated plants, or plants assessed prior to treatment. This ambiguity undermines the interpretation of the findings. It would also improve the clarity of the manuscript to align the sequence of result presentation with the order of the methods section.

Moreover, several sections contain results-related discussions or unrelated background information, which disrupts the logical flow and should be relocated accordingly. The Discussion itself suffers from a lack of structure and fails to integrate the obtained results effectively with findings from previous studies. Notably, the section spanning lines 510–541 resembles an introduction rather than a critical discussion of the current findings. In some instances, conclusions drawn by the authors appear speculative and are not directly supported by their data. For example, lines 439–442 discuss mechanisms derived from studies on other heavy metals rather than arsenic, and the role of MIPs is hypothesized without quantifying arsenic accumulation in the tissues studied (line 489). Such conclusions should be presented more carefully, clearly framed as hypotheses, and not overstated.

The Materials and Methods section must be improved with the inclusion of essential experimental details, such as treatment application protocols, exact timing of sampling, and a full description of the RNA-seq procedure, which is a critical omission.

Finally, the Conclusion section should be strengthened by identifying which of the genes discovered are most likely to be involved in arsenic detoxification pathways, based on the findings of the study.

In conclusion, while the study addresses a scientifically relevant topic and offers potentially valuable insights, significant revisions are necessary to enhance the manuscript’s clarity, logical structure, and scientific rigor.

Specific comments

Abstract: it requires revision to improve its clarity and completeness. In particular, it should include a brief presentation of the RNA-seq results, as these constitute a significant part of the study. Additionally, the experimental setup involving arsenic should be introduced more clearly, including specific information on the concentrations used (see lines 21–22).

Line 13: ‘in special’ do you mean ‘and special’?

Line 24: the statement “Permeability to AsIII was observed in both shoot and root…” appears speculative. Since arsenic content was not directly measured in the plant tissues, permeability to AsIII cannot be confirmed and should only be proposed as a hypothesis. What was actually observed is the differential expression of certain MIP genes, which may be associated with AsIII uptake. This interpretation should be presented more cautiously and clearly distinguished from direct evidence of arsenic permeability.

Keywords: add ‘Abiotic stress’

Introduction: the narrative currently lacks a coherent logical structure, with redundant and fragmented information dispersed across the section. A more effective and logical arrangement of the content could significantly improve the clarity of the Introduction. One possible reorganization is as follows:

-Begin with lines 52–54, introducing the discovery of MIPs in bacteria and humans.

-Follow with lines 71–73, which continue the narrative with the identification of other MIPs in different organisms, including plants.

-Then, integrate lines 54–56 with lines 61–63 to describe the number of MIPs identified in various plant species.

-Continue with lines 73–76, describing the conserved protein structure of MIPs (note: lines 56–58 could be removed to avoid redundancy).

-Proceed with lines 64–66, outlining the different types of MIPs, followed by lines 82–85 on the roles of various MIP subfamilies.

-Then include lines 67–70, introducing aquaporins in plants, and lines 58–61, which elaborate on their functional roles.

-Finally, present lines 80–82 to address the relationship between MIPs and As(III) in plants, followed by lines 76–79, which draw parallels with similar characteristics observed in other organisms.

-Then continue from lines 86 forward.

This suggested structure would provide a more logical and progressive flow of information, from the general discovery of MIPs to their classification, functional roles, and specific involvement in As(III) transport in plants, thereby enhancing the overall readability and coherence of the Introduction.

Lines 101-106: describe the experiment more clearly also providing the aim and importance of the study.

Results: the main concern regards to subsection 2.5 (Effect of heavy metals on chlorophyll, nitrogen content, and root length in pepper), where the experimental control is not clearly defined. A graph is presented with a control group labeled 'Before treatment,' represented by plants sampled prior to treatment, which are then compared to treated plants. However, in lines 224 and 231-232, the control results are absent from Figure 7, suggesting the presence of an additional control group in the experiment (perhaps plants that were not treated but sampled concurrently with the treated plants?). Regardless, this data should be included, as it is crucial to compare treated plants with untreated ones, rather than with plants sampled before treatment." In addition, the sampling time after treatment with AsIII should be specified (lines 224, 228, 231).

Also in RNAseq and RT-PCR controls are not clearly defined, whether they are plants sampled before treatment or not-treated plants sampled concurrently to treated plants.

Lines 298-305: unrelated background information

Discussion: this section fails to adequately contextualize the results in relation to previous research findings, resembling more of an introduction than a discussion. Furthermore, some results are presented that were not presented in the result section

Section 3.1: author’s findings on pepper are missing

Section 3.4: what is the connection between arsenic and antimony (Sb)? How does the effect of Sb relate to the findings observed in your study?

Section 3.5: the statement “Pepper plant treated with As(III) for long periods of time (24h) in shoots resulted in more permeability of MIPs in both lower and higher’ is merely speculative. Since arsenic content was not directly measured in the plant tissues, permeability to AsIII cannot be confirmed and should only be proposed as a hypothesis, based on the expression analysis of MIPs.

Lines 466-475: it is unclear what your own conclusions are.

Line 489: the statement ‘In response to a higher As(III) concentration for 24h treatment, some CATIP, CANIP and few CAPIP were up-regulated showing transport of As(III) in roots, while many CAPIP and CATIP and CAPNIP were down-regulated showing restricted uptake as a detoxification measure’ is a speculation. In fact, no data on the As(III) content in the analyzed tissues is presented by the author. Such conclusions should be presented more carefully, clearly framed as hypotheses, and not overstated.

Lines 510-541: It appears more like an introduction, where the work of other studies is presented without engaging with your own results, and no conclusions are drawn.

Furthermore, some results are introduced that were not presented in the Results section (lines 366-371).

Materials and methods: this section must be improved with additional information to improve clarity and reproducibility of the experiments. Most importantly, the procedure of RNAseq must be provided.

Line 550: ‘while MIP encoding gene sequences respectively’ not clear.

Line 598: Provide more details regarding the source of the seeds, including where they were purchased or obtained.

Lines 607-610: Provide a more detailed description of how the treatment was performed (e.g., was As(III) added to the solution?), including the duration of the treatment.

Section 4.7: provide more details about the time of sampling for both chlorophyll and root length analyses.

Section 4.8: also root samples were included in the analysis

Reviewer 2 Report

Comments and Suggestions for Authors

The article “Genomic survey of genes encoding Major intrinsic proteins (MIPs) and their response to Arsenite stress in Pepper (Capsicum annum)” is devoted to a comprehensive study of genes of the MIPs family, which provide transport of vital substances across cell membranes. The paper examines the features of MIP gene expression, as well as the morphological features of plants exposed to arsenic. The work may be interesting and relevant for a wide range of researchers - for those who study the interaction of plants with toxic metals, study MIP genes and are looking for ways to reduce the harm from toxic metals to plants using genetic engineering.

However, a number of shortcomings were noted in the work. I believe that one of the main shortcomings is the obvious imbalance between the data obtained theoretically and based on someone else's experiment (Kim S. et al., Nature Genetics 2014, 46, 270–785) and the experimental data obtained during the experiment conducted by the authors of the manuscript themselves. The second significant shortcoming is the Discussion section, which needs to be shortened and redone.

Remarks:

  1. Lines 62, 63, 356, 531, 551, etc. – Latin names should be written in italics; after the first use of the full Latin name of a species in the main text of the article, it should be written in abbreviated form (and not as, for example, on line 551).
  2. Lines 90-92 are not a coherent sentence, or something is missing.
  3. Lines 96-98 are not coherent
  4. Line 120 is a typo Ip
  5. Figures 1, 4, 6 and 9 are very small, even at the highest magnification there is no adequate clarity.
  6. The Discussion section should not be an extended summary of the results, and there is no need to justify any particular method used. The Discussion should contain an interpretation of the results of the study and their significance in the context of the existing scientific literature. It is the interpretation of the results that is missing from the current Discussion. In general, the Discussion section usually contains: a) a brief overview of the most significant results obtained during the study; b) an explanation of what the results mean and how they relate to previous studies; c) a discussion of how the results of your study relate to the data of other authors, identifying agreements and disagreements; d) an indication of possible limitations of your study that may affect the interpretation of the results; e) an explanation of any unexpected findings or results that do not meet expectations.
  7. Line 432 – perhaps the reference here is incorrectly formatted?
  8. Half of section 3.4 is devoted to describing and listing the effects of Sb on plants without discussing or comparing it with the results obtained in this study. An explanation of why so much attention is given to antimony in this section should be at the beginning of the section, not at the end.
  9. I It should be more clearly stated that the data reflected in Figures 8 and 9 were obtained in different experiments, only one of which was conducted during this study.

Round 2

Reviewer 1 Report

Comments and Suggestions for Authors

I acknowledge the authors’ effort in revising the manuscript according to the reviewers’ suggestions. The overall structure and clarity of the manuscript have improved, and the revisions are generally appropriate and contribute positively to the quality of the work. However, there are still several important issues that need to be addressed, particularly in the Discussion section.

Although this section has been reorganized, it still lacks clarity in several parts. In particular, paragraphs 3.3 and 3.4 (lines 408–415), as well as paragraph 3.5 (lines 434–435, 441–443, and 462–465), remain difficult to follow. The reasoning lacks logical flow, and the English language is, at times, imprecise or awkward, which further hampers the reader's understanding. The arguments often appear as a disjointed list of statements rather than a coherent discussion that builds on the results and places them in context. This undermines the purpose of the Discussion section, which should provide a critical interpretation of the findings and link them to the existing body of knowledge.

Moreover, the information presented in lines 494–506 seems disconnected from the rest of the discussion. It is unclear why these details are included, as they do not appear to contribute to the interpretation of the results or suggest any meaningful application. The authors are encouraged to either clarify the relevance of this passage or consider removing it.

I recommend that the authors revise the Discussion more thoroughly, focusing on improving the clarity, logical flow, and coherence of the argumentation. Attention to language polishing and ensuring that each paragraph contributes meaningfully to interpreting the results would significantly enhance the manuscript.

More specific comments and suggestions for text editing are provided in the attached PDF.

Reviewer 2 Report

Comments and Suggestions for Authors

All comments made earlier by the authors have been taken into account. The manuscript can be recommended for further preparation for publication in the journal.
